# Gene-Environment Interactions in Vitamin D Status and Sun Exposure: A Systematic Review with Recommendations for Future Research

**DOI:** 10.3390/nu14132735

**Published:** 2022-06-30

**Authors:** Rasha Shraim, Conor MacDonnchadha, Lauren Vrbanic, Ross McManus, Lina Zgaga

**Affiliations:** 1Department of Public Health and Primary Care, Institute of Population Health, Trinity College Dublin, D24 DH74 Dublin, Ireland; rshraim@tcd.ie (R.S.); macdonco@tcd.ie (C.M.); vrbanicl@tcd.ie (L.V.); 2Department of Clinical Medicine, Trinity Translational Medicine Institute, Trinity College Dublin, D08 W9RT Dublin, Ireland; rmcmanus@tcd.ie; 3The SFI Centre for Research Training in Genomics Data Sciences, National University of Ireland Galway, H91 CF50 Galway, Ireland

**Keywords:** vitamin D, gene-environment interaction, sun exposure

## Abstract

Vitamin D is essential for good health. Dermal vitamin D production is dependent on environmental factors such as season and latitude, and personal factors such as time spent outdoors and genetics. Varying heritability of vitamin D status by season has been reported, suggesting that gene-environment interactions (GxE) may play a key role. Thus, understanding GxE might significantly improve our understanding of determinants of vitamin D status. The objective of this review was to survey the existing methods in GxE on vitamin D studies and report on GxE effect estimates. We searched the Embase, Medline (Ovid), and Web of Science (Core Collection) databases. We included only primary research that reported on GxE effects on vitamin D status using 25-hydroxyvitamin D as a biomarker. Sun exposure was the only environmental exposure identified in these studies. The quality assessment followed the Newcastle–Ottawa Scale for cohort studies. Seven studies were included in the final narrative synthesis. We evaluate the limitations and findings of the available GxE in vitamin D research and provide recommendations for future GxE research. The systematic review was registered on PROSPERO (CRD42021238081).

## 1. Introduction

Adequate vitamin D status is essential for good health. Vitamin D deficiency has been linked with the risk of a range of diseases including bone health, cancer, autoimmune disorders, cardiovascular diseases, and many others [1,2], including mortality [3]. Given its significance, vitamin D deficiency is a notable public health concern worldwide [4].

Production in the skin following sun exposure accounts for approximately 80% or more of vitamin D in humans on average, with the remainder ingested in the form of food or supplements. Dermal vitamin D production is dependent on environmental factors (e.g., season and latitude), and personal factors such as skin type and surface area exposed, including genetic make-up [5,6,7]. Figure 1 illustrates the vitamin D pathway. When the skin is exposed to ultraviolet B (UVB) radiation, in a photochemical reaction 7-dehydrocholesterol (7-DHC) is converted to previtamin $D_{3}$, which spontaneously transforms into vitamin $D_{3}$ (cholecalciferol). Vitamin $D_{3}$ is next processed in the liver to 25-hydroxyvitamin D (25(OH)D), the major circulating form and the standard biomarker of vitamin D status [5,8].

Interestingly, among the first genetic variants to be associated with vitamin D status was an eQTL variant associated with altered expression of the 7-Dehydrocholesterol [DHC] Reductase (DHCR7) gene [9]. DHCR7 encodes an enzyme that catalyses the conversion of 7-DHC to cholesterol, thereby directly affecting the availability of a key vitamin D precursor, 7-DHC. This illustrates a clear example in vitamin D metabolism where genetic and environmental factors intersect, providing a strong rationale for gene-environment (GxE) interaction analysis (see [10] for a detailed discussion of GxE in epidemiology).

In addition to UVB exposure and genetic background, serum 25(OH)D levels are influenced by many demographic and lifestyle factors, such as diet, BMI, and age [11,12]. However, twin studies have shown that up to 70% of the variability in 25(OH)D is explained by genetic factors, indicating that it is a highly heritable trait [13,14,15]. Jiang et al. [15] reported heritability estimates from 20 to 90%. This variability may be explained by the influence of season [13,14]. Some studies report varying heritability between the winter and the summer, when dermal production is likely the dominant source [12,13,14,16,17]. This suggests a modification of genetic effect by season (i.e., intensity of solar radiation), where the impact of genetic effects is more pronounced in the winter. Furthermore, while genome-wide association studies (GWAS) have so far identified over 140 independent loci associated with vitamin D level, known SNP heritability still accounts for less than 20% of the variance, leaving a large proportion of vitamin D status heritability unexplained [12,16,18]. GxE interactions may be key to further understanding the heritability of vitamin D status and identifying the modifiable risk factors of vitamin D deficiency. Studying GxE interactions can elucidate the ‘population-attributable’ effects of environmental exposures, such as UVB, on the biological pathways of vitamin D metabolism and allow us to tailor public health advice to the population or individual’s genetics [19].

We hypothesise that gene-environment interactions (GxE) play a major role in vitamin D status and that uncovering these interactions will improve our understanding of vitamin D level determinants (see [20]). In this report, we conduct a systematic review of the available literature on GxE in vitamin D status. The objective of this review was to survey the existing methods and effect estimates in GxE studies of vitamin D, especially relating to sun exposure.

## 2. Materials and Methods

### 2.1. Search Strategy

This systematic review was conducted according to the Preferred Reporting Items for Systematic reviews and Meta-Analyses (PRISMA) statement [21]. The protocol was registered with PROSPERO (CRD42021238081).

We searched the Embase, Medline (Ovid), and Web of Science (Core Collection) databases without any language restrictions from inception until 15 June 2022. The key search terms were ‘vitamin D’ and ‘gene-environment interactions’. The full search strategy for each database is available in Appendix A.

### 2.2. Inclusion and Exclusion Criteria

Studies were included if they examined participants from the general population, without specific illnesses. Any observational study (cohort, case-control, cross-sectional) or randomised controlled trials were eligible, as long as they included an analysis of gene-environment interactions. The primary outcome was vitamin D level, measured as circulating 25(OH)D concentration. The intervention/exposure of interest was gene-environment interaction, including only natural environmental factors (primarily sun exposure). We excluded studies that only reported on genetic factors or environmental factors independently but not their interaction, only examined the effect of human intervention (including diet and supplements) on vitamin D level, or only examined the effect of vitamin D level on other outcomes.

### 2.3. Study Selection

Covidence [22] was used to aid study selection and data extraction. After removal of duplicates, titles and abstracts were screened by two independent researchers (LV and CD). Disagreements were resolved by discussion with a third researcher (RS). Full texts were obtained and screened for eligible studies by RS. Reviews were excluded but their reference lists were screened, as were those from eligible studies.

### 2.4. Data Extraction and Quality Assessment

For each eligible study, two researchers (LZ and RS) independently extracted and recorded the following data in Covidence: publication characteristics (including author, journal, publication date), study details (including design, type, aim, sample size, population characteristics, environmental exposure measures), and outcome measures (including vitamin D levels, effect size estimates). Quality assessment was performed following the Newcastle–Ottawa Scale for cohort studies (all of the included studies were cohort studies; ref. [23]). By assessing the GxE pairs studied to date, we found no two studies that investigated the same interaction. Thus, we were unable to conduct a meta-analysis of GxE interaction effects. Nonetheless, we summarise and present the effects of GxE on 25(OH)D reported in the literature and describe the different approaches used to study them, including the various genetic variants, environmental proxies, and effect measures used to date. The pooled mean 25(OH)D concentration across the included studies was assessed using the R package *meta* [24]. Given the considerable between-study heterogeneity, a random-effects model was used to pool 25(OH)D estimates. The restricted maximum likelihood estimator was used to calculate the heterogeneity variance $\tau^{2}$ and the Knapp–Hartung adjustments to calculate the confidence interval around the pooled mean [24,25]. Two studies were conducted using the same cohort, the UK Biobank. We present both, but when synthesising results we keep Revez et al. [16] (N = 318,851) and exclude the smaller one, Manousaki et al. [12] (N = 193,809).

## 3. Results

### 3.1. Study Selection and Characteristics

We identified 783 records. Figure 2 outlines the search process. We included 7 studies in the final synthesis. Overall, these studies comprised 332,418 participants (Table 1). The sample size varied greatly, from 504 in Robien et al. [26] to 318,851 in Revez et al. [16]. Some studies included Asian and African individuals, but the majority of participants were white European and approximately half were female, except in Shao et al. [27] and Engelman et al. [28] which included only females. The pooled mean 25(OH)D concentration was 55.6 nmol/L (95% CI 44.8–69.1, Figure 3). Heterogeneity between studies was very high ($I^{2}>99\%$; heterogeneity variance $\tau^{2}=0.0422$, 95% CI:0.02–0.26, log-transformed).

### 3.2. Risk of Bias in Studies

All of the included papers had an overall low risk of bias score. Shao et al. and Engelman et al. [27,28] had a potential risk of bias in the representativeness of the cohort given that they sampled pregnant and post-menopausal women, respectively.

### 3.3. Research Design and Study Samples

A summary of study designs and findings is presented in Table 1. The approach to the GxE analysis varied greatly.

### 3.4. Environmental Exposure

Four of the studies used season as a proxy for sunlight exposure [12,16,27,28], two used the number of hours of sunlight exposure [26,29], and one calculated exposure based on independent UV data [30].

### 3.5. Genetic Factors

There was a variety among examined genetic factors, in terms of their type, where some studies reported single-nucleotide polymorphisms (SNP) [12,16,27,28,29], others haplotypes [26], and polygenic risk scores (PGS) [30]. The interaction analysis was usually limited to genetic variants found to be significantly associated with vitamin D status independently. Shao et al. [27] was the only study that reported on effect size per SNP genotype. The only SNPs examined in two studies were rs2060793, rs10500804, rs11023380, and rs11023374 in the CYPR21 gene [16,28]. The GC and CYPR21 genes were included in four studies [12,16,26,28] but effect size was measured for different SNPs (rs4588 was in $r^{2}>0.9$ LD with rs11723621 [12] and rs1352846 [16], Appendix A).

### 3.6. Covariates

The choice of covariates also varied. All studies adjusted for age, sex, and vitamin D intake, and some additionally adjusted for one or more of: BMI, season, ethnicity, assessment centre, socieconomic markers, or physical activity.

### 3.7. Interaction Findings

Apart from [26], all included studies found significant evidence for GxE, but the direction of the effect was not always clear. We present a summary below and in Figure 4 on some of the key genes included in these studies [31]. Overview of genetic factors, including SNP selection and significance thresholds, is presented in Table 2.

### 3.8. Interaction in GC

GC encodes a vitamin D binding protein (also known as Group-specific Component) that binds to and transports vitamin D and was examined in 4 studies ([12,16,26,28]. Robien et al. [26] did not find evidence for interaction between the GC haplotype and hours spent sitting at work (*p*-value = 0.24). However, Engelman et al. [28] reported a significant interaction (*p*-value = 0.01) for rs7041, a SNP within GC, where the b-coefficient for the high-exposure group was almost sixteen times that in the low-exposure group (stratified by winter/summer, −0.33 vs. −0.02, respectively). There was no evidence for interaction in the other GC SNP, rs4588 (*p*-value = 0.17, ref. [28]). Neither Revez et al. [16] nor Manousaki et al. [12] found evidence for interaction in their selected SNPs in this gene (corrected $p>5\times 10^{-8}$, $p>3.6\times 10^{-4}$, respectively), including rs4588 (to which rs1352846 from [16] and rs11723621 from [12] are in complete LD (r2 and D’ > 0.99, ref. [32]).

### 3.9. Interaction in VDR

VDR (vitamin D receptor gene) encodes the vitamin $D_{3}$ receptor. While several of the studies reported on VDR in their main analyses, only Livingstone et al. [29] included VDR in their interaction analysis. Livingstone et al. [29] found the relationship between rs2228570 and 25(OH)D to be modulated by the time spent in the sunlight during the week (*p*-value = 0.009) and total sunlight exposure (weekdays plus weekend days, *p* = 0.045).

### 3.10. Interaction in CYP2R1

CYP2R1 encodes an enzyme that converts vitamin $D_{3}$ into 25(OH)D. Engelman et al. [28] did not find significant evidence for interaction in any of the 4 SNPs examined (Table 2), whereas Revez et al. [16] found evidence for interaction in all 4 SNPs at genome-wide significance level (rs2060793, rs10500804, rs11023380, and rs11023374). Manousaki et al. [12] reported significant interaction for several SNPs within CYP2R1 at Bonferroni-corrected threshold. While none overlapped, some were in LD with SNPs from Engelman et al. [28] (Appendix A).

### 3.11. Interaction with Other Genetic Variants

Manousaki et al. [12] also reported a strong interaction in the rs8018720 variant in the SEC23A locus. Revez et al. [16] found some evidence for interaction with season in 1127 variants in chromosomes 7, 11, 14, and 15. Shao et al. [27] observed evidence for interaction with season in two other cytochrome genes, CYP27A1 and CYP3A4 (*p*-value 0.02 and 0.004, respectively). Lastly, Hatchell et al. [30] examined interaction using a PGS approach. The ancestry-specific PGS was calculated based on the results from a multi-ethnic vitamin D GWAS meta-analysis [33,34] They observed significant interaction between PGS and UV in the month of blood draw in their European ancestry sample but not in their African ancestry sample (*p*-value 0.021 and 0.71, respectively; ref. [30]).

## 4. Discussion

In this systematic review, we report and summarise evidence of GxE interaction on vitamin D status. The findings suggest that the effect of genotype was modulated by sunlight exposure, whether using individual behaviour [28,29], or season [12,16,28,30]. While we were unable to derive a precise estimate of GxE effect on vitamin D status due to the small number of studies included, the available research does suggest that the interaction between genes and sun exposure may be the key to better understanding determinants of vitamin D status. The case for this is further strengthened by the fact that GxE interactions reported to date have predominantly been found in genes with clear relevance for vitamin D status and metabolism such as VDR (vitamin D receptor), GC (vitamin D binding protein), SEC23A (protein transport), CYP2R1 (vitamin $D_{3}$ enzyme), and CYP27A1 (cholestrol metabolism). Careful consideration of these studies, and in particular the Robien et al. [26] study—the only study where no significant GxE interaction was found—yielded important notions for future research.

### 4.1. Sample Size and Power

Median sample size of the included studies was only 1,258. This is quite small in the context of genetic research and the expected effect sizes, and thus likely underpowered [10]. Non-significant interactions should be considered with caution. Here, only the smallest of the included studies (N = 504) did not find any evidence of GxE interaction [26]. For example, in the CYP2R1 gene the interaction *p*-value was not significant in Engelman et al. [28] for any of the four evaluated SNPs whereas the GWS *p*-value was significant for all four in Revez et al. [16]. However, mega-cohorts such as the UK Biobank are not abundant, so a collaborative approach to GxE studies will likely be essential to uncovering GxE interactions affecting human traits. This can be achieved through large international consortia, or alternatively by meta-analysis of independently published research. In Table 3 we summarise our recommendations that would enable meta-analyses in the future.

### 4.2. Variability in Exposures

Robien et al. [26] study was performed in Singapore (latitude: 1°17${}^{\prime }$ N). Geographical regions located close to the equator experience the smallest seasonal variation in daily UV dose because distance to the sun remains largely unchanged throughout the year (albeit seasonal impact of weather might play a role). Restricted variability in environmental exposure limits the capability of a study to find significant associations. Taken to the extreme, in a population where environmental exposure is the same for all of the participants, detecting associations is precluded at the outset—even if associations exist [35]. Therefore, studies conducted in regions where UV variability is greater and studies that recruit participants over a year or longer (capturing the broadest range in exposure at a given location) are better placed for this research. There was considerable variety in the selected genetic factors. Some studies opted for whole genome or PGS while others reported on specific vitamin D genes or SNPs. Allele or genotype frequencies capture the variability of genetic factors (i.e., similarly to seasonal UV variability throughout the year, alleles represent the genetic variability within a population). Depending on the sample size, low frequency or rare alleles may diminish the power to observe associations. Thus, population-specific allele frequencies and chosen genetic quality control cut-offs (e.g., MAF cut-offs) may impact substantially on study findings and need to be clearly reported (see comparison of Manousaki et al. [12] and Revez et al. [16] below).

### 4.3. Choice of Environmental Exposure

The choice of environmental exposure measures used to approximate sun exposure differed notably. Four of the included studies reported on the season of blood draw, three used personal reports of sun exposure (including the number of hours spent “doing vigorous work” and “sitting at work”), and one included independent UV radiation data. In some cases, measures of physical activity are used to approximate exposure to sunshine [26], assuming the activity is performed outdoors but this may not always be the case. UV dose varies dramatically by latitude, altitude, weather and other factors. While the season of blood draw can easily be obtained and is unbiased, the actual UV dose will vary significantly depending on the location during any given season.For example, the average UVB dose during the month of April is approximately 75 mJ/cm${}^{2}$ in Aberdeen, Scotland, and 120 mJ/cm${}^{2}$ in Plymouth, England [36]. Most studies dichotomised the season of blood draw into summer and winter. In the UK, the daily median UV dose in autumn was almost eight times that in winter [36]. Therefore, two individuals from the UK who are both labelled as “winter” may have UVB dose that is 8-fold different. Because of the large UVB dose variability within any given season, and because of the large UVB dose variability between locations but during the same time of the year, season of blood draw is a poor proxy of UVB exposure and a significant amount of information is lost, rendering a noisy and imprecise environmental exposure variable, that is not comparable across different locations. For these reasons, we recommend that researchers incorporate independent UV data based on the date of blood draw. UV data are increasingly available through several public or research resources such as NASA or Google Earth. Finally, vitamin D level will be affected by the actual exposure to ambient UV, determined by individual behaviour such as indoor- versus outdoor-oriented lifestyle, clothing, or sun holidays (e.g., [37,38]) but this is more challenging—if not impossible—to capture accurately and standardise. The ‘season’ variable may still be useful as a covariate when the environmental exposure is calculated from independent UV data. The effect of season may relate to individuals’ behaviour patterns across the year. For example, if people are more likely to spend time outside during the summer than the winter then the difference between the environmental exposure ‘summer’ and ‘winter’ as discussed above is even further compounded by the increased time spent outdoors. The usefulness of this variable will depend on the geographical location of the study and data availability (e.g., availability of a “time spent outdoors” variable) but it should be evaluated where possible.

### 4.4. Significance Thresholds

Five studies included in this review considered *p* < 0.05 to constitute a significant finding, and two used a Bonferroni-corrected *p*-value ($<3.6\times 10^{-4}$[12] and $<5\times 10^{-8}$ [16]). Significance thresholds should be adjusted when testing multiple SNPs. When SNPs within the same gene or in high LD are investigated, the Bonferroni correction might be too stringent (the SNPs are not independent). Cut-offs for statistical significance should be considered carefully and interpreted within the study context. Where appropriate, other indicators such as false-discovery rate should be applied.

### 4.5. Methodological Implications

Methodological approaches can have a major impact on results and need to be clearly documented. For instance, Manousaki et al. [12] and Revez et al. [16] both examined the European ancestry population in the UK Biobank cohort but due to methodological differences, their reported results were somewhat different. They both used season of blood draw as the environmental exposure proxy, as a dichotomous winter/summer variable, although the exact months were slightly different (winter defined as January–March in [12] and December–April in [16]). Revez et al. [16] tested over 6 million variants in 318,851 individuals initially and identified 1127 variants with a genome-wide significant interaction (*p* $<5\times 10^{-8}$, ref. [16]). Manousaki et al. [12] tested 138 SNPs they found to be significantly associated with vitamin D status in 193,809 individuals and reported significant evidence for interaction for 12 independent SNPs. There were no overlapping interaction SNPs among those found significant by Manousaki et al. [12] and Revez et al. [16]; and where Manousaki et al. [12] found the interaction effect (β genotype*season) of the 12 significant SNPs to be in the same direction as that of the β on 25(OH)D levels, Revez et al. [16] found the effect was reversed for two of their five SNPs.

Broadly, GxE research on vitamin D status would benefit from following similar guidelines as those outlined in Dunn et al. [39] on GxE in youth depression. We summarize some of their recommendations and include recommendations specific to vitamin D research below (Table 3; ref. [39]).

GxE studies should report clearly on the chosen model and its parameters, the interaction term, effect size measurements, and significance thresholds. Researchers can consult guidelines such as STROBE (STrengthening the REporting of OBservational studies in Epidemiology) and STREGA (Strengthening The Reporting of Genetic Association studies) when reporting methods and results [40,41]. Age, sex and BMI and vitamin D supplementation are essential covariates that should be evaluated. Additionally, skin colour influences the production of vitamin D, with dark skin less efficiently producing vitamin D than white skin [42]. Therefore, a measure of skin pigmentation or colour (e.g., Fitzpatrick skin type) should also be evaluated. None of the included studies incorporated skin colour variables. Limited information was available in the included studies on the model and details of the interaction analysis. Researchers should report their model assumptions, including the assumption of an additive or multiplicative interaction model, the model parameters and interaction terms, and whether a statistical comparison was made with and without the interaction term. Reporting main effects in addition to the interaction effect would support the interpretation and understanding of the GxE analysis, clarifying whether an interaction is likely to be biologically relevant or only statistically significant. Researchers should always report their effect estimates as well as statistical significance to enable comparisons across studies.

The small number of studies identified reflects the current state of interaction research in vitamin D. While sun exposure accounts for most vitamin D in humans [5], the interplay of genetics and environment in determining vitamin D levels remains understudied. For instance, Revez et al. [16] identified at least 10 vQTL loci with no significant seasonal interaction, suggesting the presence of other GxE with yet to be identified environmental exposures. Only one of the studies included set out to examine GxE as a primary goal, most included GxE as a secondary analysis. Vitamin D deficiency is considered a global public health concern [4], and a better understanding of the interplay between genetics and environmental conditions is needed to address this public health issue properly. All of the authors of the included studies recommend a more integrated approach to the study of the underlying mechanisms determining vitamin D status [12,16,26,27,28,29,30].

Notably, the majority of the excluded studies were excluded because they reported on gene-environment interactions with vitamin D level being the ‘environmental exposure’ of interest. Broadly, epidemiology research is shifting from the candidate gene or candidate risk/environmental factor to the genome-wide and GxE approach [10,19,20]. Some of the common outcomes in these studies were multiple sclerosis, bone health, inflammation, and other autoimmune illnesses. Given the expanding interest in vitamin D [43,44], epidemiology research would benefit greatly from understanding first which factors influence vitamin D status itself. Interaction analysis can help dissect the effect of complex exposures on complex traits and explain the heterogeneity observed in different vitamin D studies [10].

### 4.6. Strengths and Limitations

We systematically surveyed the available research and provided moderate evidence of interaction in circulating vitamin D concentration. Records were assessed and data were extracted by two independent reviewers, which minimised the risk of excluding relevant papers or information. One limitation of this review is that we could not carry out a meta-analysis due to the heterogeneity across studies. One important limitation of the research field itself is the availability of data in non-European ancestry populations, who may have different determinants of vitamin D status. While half of the studies reported on Asian or African ancestry individuals, they only made up 0.7% of the total sample size. The generalizability of such results is limited and it would be beneficial to gather evidence from more diverse populations, both ethnically and geographically. Another common limitation is the use of poor proxy measures for environmental exposure (discussed above).

## 5. Conclusions

This review summarises the available research on the role of GxE interactions in determining vitamin D status. The available evidence suggests the presence of GxE interactions, although further investigation is needed to fully understand the role of GxE in the vitamin D pathway. The recommendations provided aim to improve our approach to the study of GxE and allow comparison and meta-analysis across studies.

## Figures and Tables

**Figure 1 nutrients-14-02735-f001:**
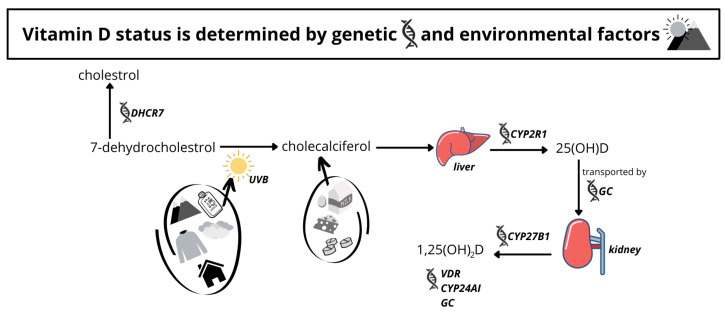
A schematic of the vitamin D metabolic pathway. Vitamin D status is affected by genes such as DHCR7 (7-Dehydrocholesterol Reductase) and GC (Group Component) as well as environmental and personal factors such as climate, clothing, and supplement use. 25(OH)D: 25-hydroxyvitamin D; 1,25(OH)2D: 1,25-dihydroxyvitamin D; UVB: ultraviolet B radiation; VDR: vitamin D receptor.

**Figure 2 nutrients-14-02735-f002:**
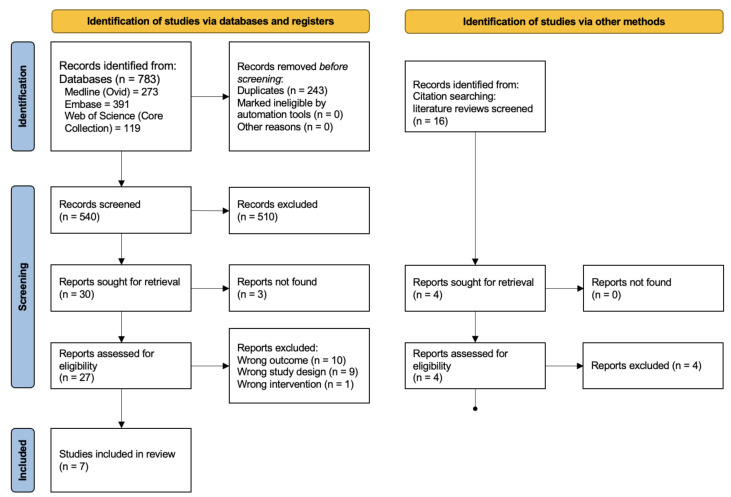
Systematic review study selection PRISMA flow diagram.

**Figure 3 nutrients-14-02735-f003:**
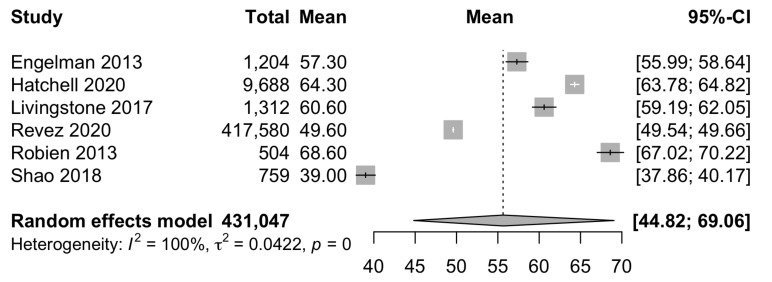
Pooled mean of 25(OH)D across the included studies [26,27,28,29,30]. Revez et al. [16] and Manousaki et al. [12] use the same population, the latter was excluded (see Methods for details).

**Figure 4 nutrients-14-02735-f004:**
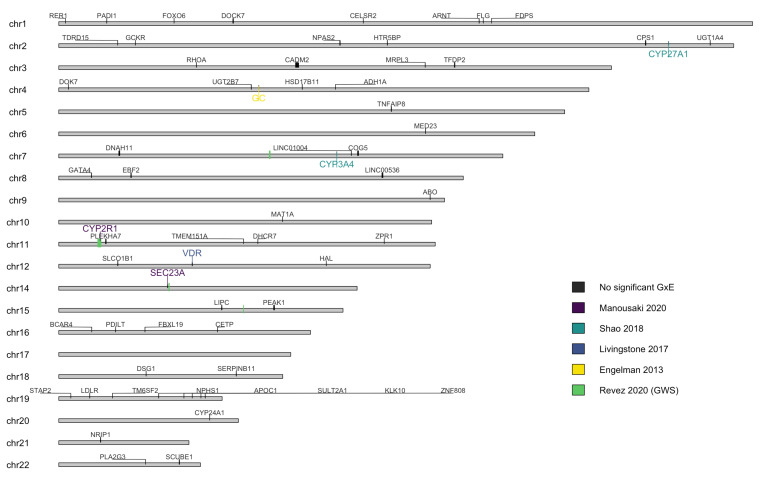
Chromosomal map showing the genes assessed for gene-environment interactions from the studies included in this review [12,27,28,30]. While most of the studies used SNPs, only the genes that those SNPs fall within are shown, for easier visualisation. Revez et al. [16] identified over 1000 SNPs with significant GWS GxE, they are shown as regions on the chromosomes. Hatchell et al. [33] Legend: genes in colour were found to have significant GxE interactions, genes in grey were analysed but not found to have significant interactions in the included studies. Note: chromosomes 13, X, and Y are not shown as they were not included in any of the studies.

**Table 1 nutrients-14-02735-t001:** Summary of the characteristics of the included studies on gene-environment interactions in vitamin D.

Study	Sample Characteristics (GxE)	Mean 25(OH)D Concentration	Gene (G) (Quality Control)	Environment (E)	Other Covariates	GxE Findings
Engelman 2013 [28]	1204 postmenopausal European women age 50–79 recruited in the US 1993–1998 (sampled from CAREDS eye disease cohort)	Dec–May: 50.1 (SD 22.1), Jun–Nov: 63.3 (SD 22.7) nmol/L (chemiluminescence or radioimmunoassay)	SNPs in GC, DHCR7, CYP2R1, and CYP24A1 (HWE, MAF, call rate, heterozygosity, concordance rate)	season of blood draw: winter/spring (Dec–May) and summer/fall (June–Nov); individual sun exposure: weekly duration of total recreational physical activity and yard work; time spent in direct sunglight at baseline: a sunlight exposure questionnaire administered at baseline (2001–2004)	vitamin D intake, waist circumference, season of blood draw, total cholesterol, and hours in sunlight	SNP × season: interaction was significant for only one gene-environment pair (rs7041- season; *p* = 0.01), where the b-coefficient for the high-exposure group was much more than twice that in the low-exposure group (−0.33 vs. −0.02, respectively); interaction between genetic risk score and external source of vitamin D was significant for season of blood draw (*p* = 0.04) but not for vitamin D intake (*p* = 0.26)
Robien 2013 [26]	504 government-built housing estate residents with Hokkien or Cantonese dialect age 55.7 (7.8), 56% F, recruited in Singapore 1993–1998	68.6 nmol/L (SD 18.3) (chemiluminescence immunoassay)	GC haplotype (HWE, MAF, call rate)	average number of hours spent sitting at work and hours spent doing vigorous work, taken as surrogates for time spent indoors and outdoors, respectively	Dialect group, education level, menopausal status (women), BMI, height, weight, body surface area, physical activity, smoking status, hours spent sitting at work, season of blood draw, use of cod liver oil supplements and dietary intake of vitamin D, Ca, fish, dairy products and alcohol	GC haplotype × hours spent sitting at work: p-interaction = 0.24 (not significant)
Livingstone 2017 [29]	1312 healthy university students age 40.2 (13.0), 97% Caucasian, recruited in Ireland, the Netherlands, Greece, the UK, Poland, and Germany 2012–2013	60.6 nmol/L (SD 26.4) (chromatography)	SNPs from VDR, GC and PGS from the minor alleles of VDR and GC (HWE, LD)	Weekend and weekday sunlight exposure (during day light on a typical week day and on a weekend day during the sunny months of the year (i.e., April to September) collapsed into <20 min‚ 20 min–2 h, and >2 h (dietary intake of vitamin D: Online food frequency questionnaire (FFQ) of foods and supplements)	age, sex, BMI, ethnicity, country, season, vitamin D intake (food only) and vitamin D supplementation	SNP × sunlight exposure (and SNP × diet): The relationship between VDR rs2228570 genotype and 25(OH)D concentration was modulated by time spent in the sunlight during the week (p-interaction = 0.009). When total sunlight exposure (week- days plus weekend days) was considered, the interaction with VDR rs2228570 remained significant but evidence for the interaction was weaker (*p* = 0.045). No significant interactions were observed between genotype and dietary vitamin D
Shao 2018 [27]	759 healthy pregnant Chinese women age 28 (3), recruited in China 2011–2014 (no history of chronic or acute disease or mental disorders)	39 (SD 16.25) nmol/L (chromatography)	DHCR7, GC, CYP24A1, CYP27A1, CYP27B1, CYP2R1, CYP3A4, LRP2, NADSYN1, VDR (HWE, MAF, $r^{2}$)	season, merged into summer/fall (June-November) and winter/spring (December–May)	Age, pre-pregnancy BMI, sampling season, vitamin D supplements, physical activity	SNP × season: interactions were observed between season and CYP27A1 rs933994 (*p* = 0.02), CYP3A4 rs2246709 (*p* = 0.004); similar trends were also found in the logistic analysis of interactions between CYP27A1 rs933994 (*p* = 0.05), CYP3A4 rs2246709 (*p* = 0.03) and seasons on vitamin D deficiency (see [27] Table 8)
Hatchell 2020 [30]	9688 European and African ancestry individuals age 45–84, 59% F, 11% African, recruited in the USA 1990–2002 (sampled from Atherosclerosis cohorts and population-based cohorts)	range of 18.9 to 30.1 ng/ml (see [30] Table 1) (chemiluminescence and chromatography)	PGS, (HWE, MAF, imputation quality score, sample and SNP call rate)	continuous UV radiation based on month of blood draw and location using UV data from the National Weather Service Climate Prediction Center historical database (range: 0.7–9.5 UV index units)	age, sex, BMI, cohort, vitamin D intake, and available UV radiation (physical activity where available)	PGS × season (and PGS × vitamin D intake): in European and PGS*UV model, beta (SE) = 0.017 (0.0073) (*p*-value <0.021) (see [30] Table 2). The 2-DF PGS*intake, 1-DF PGS*UV, and 2-DF PGS*UV results were statistically significant in participants of European ancestry *p* = $3.3\times 10^{-18}$, *p* = $2.1\times 10^{-2}$, and *p* = $2.4\times 10^{-19}$, respectively). No significant interactions in African ancestry sample (limited power).
Manousaki 2020 [12]	193,809 white British individuals age 56.8 (8.0), 54.1% F recruited in the UK 2006–2010 (population-based cohort UKBB)	70.0 (SD 34.7) (chemiluminescence)	138 conditionally independent SNPs (HWE, MAF, imputation quality score)	season of measurement, winter (Jan–March), summer (July–Sept)	age, sex, season of measurement, and vitamin D supplementation (BMI excluded to avoid introducing collider bias)	SNP × season of measurement: significant interaction with season in 11 independent SNPs in the CYP2R1 locus on chromosome 11 and in a single variant in the SEC23A locus on chromosome 14 (all *p* < $3.6\times 10^{-4}$), strongest interaction was found for rs117913124 in CYP2R1 (p -interaction $1.5\times 10^{-55}$)
Revez 2020 [16]	318,851 white British individuals age 40–69 recruited in the UK 2006–2010 (population-based cohort UKBB)	median, mean and interquartile range of 47.9, 49.6, 33.5–63.2 nmol/L (chemiluminescence)	1127 genome-wide significant variants, (MAF, genome-wide significance)	season of blood draw, winter (Dec–April) and summer (June–Oct)	age, sex, (with and without BMI), genotyping batch, assessment centre, month of testing, supplement intake and thefirst four ancestry PCs	variant × season of blood draw: Of 6,098,063 variants tested (MAF > 0.05), 1127 had a GWS (*p* < $5\times 10^{-8}$) interaction with season, and 1120 (99%) were also GWS in the vQTL analysis. Of the 20 vQTL loci without significant GxE with season, at least half showed no evidence at all for GxE with season, so these variants are candidates for GxE with other environmental factors

Abbreviations: UK Biobank (UKBB), polygenic risk score (PGS), Hardy-Weinberg Equilibrium (HWE), minor allele frequency (MAF), linkage disequilibrium (LD), principal component (PC), variant quantitative trait locus (vQTL).

**Table 2 nutrients-14-02735-t002:** Genetic variants tested in each study in the main genetic analysis, the subset that was used in the GxE interaction analysis, and the significance threshold used for the latter.

Study	Main Analysis	G in GxE	GxE Significance
Robien 2013 [26]	55 SNPs in VDR, CYP2R1, CYP3A4, CYP27B1, CYP24A1, and GC	GC haplotype	*p* < 0.05
Engelman 2013 [28]	29 SNPs in GC, DHCR7, CYP2R1, and CYP24A1	GC (rs4588, rs7401) and CYPR21 (rs2060793, rs10500804, rs11023380, rs11023374)	*p* < 0.05
Livingstone 2017 [29]	5 SNPs from VDR and GC	VDR (rs2228570)	*p* < 0.05
Shao 2018 [27]	51 SNPs in NADSYN1/DHCR7, GC, CYP3A4, CYP2R1, CYP27A1, CYP27B1, VDR, CYP24A1, and LRP2	CYP27A1 (rs933994) and CYP3A4 (rs2246709) (not clear if any other snps were tested)	*p* < 0.05
Hatchell 2020 [30]	PGS	PGS	*p* < 0.05
Manousaki 2020 [12]	genome-wide (20,370,874 variants)	138 conditionally independent lead SNPs	*p* < $3.6\times 10^{-4}$, Bonferroni-corrected threshold (0.05/number of SNPs)
Revez 2020 [16]	genome-wide (8,806,780 SNPs GWAS, MAF > 0.01)	6,098,063 variants (MAF > 0.05)	*p* < $5\times 10^{-8}$, genome-wide significance

Genes: Vitamin D receptor (VDR), Group-specific component (GC), Cytochrome P450 Family 2 Subfamily R Member 1 (CYP2R1), Cytochrome P450 Family 3 Subfamily A Member 4 (CYP3A4), Cytochrome P450 Family 27 Subfamily B Member 1 (CYP27B1), Cytochrome P450 Family 24 Subfamily A Member 1 (CYP24A1), 7- Dehydrocholesterol Reductase (DHCR7), NAD Synthetase 1 (NADSYN1), LDL Receptor Related Protein 2 (LRP2).

**Table 3 nutrients-14-02735-t003:** Recommendations for GxE studies on Vitamin D.

Vitamin D	Specify which vitamin D measure was used (e.g., 25(OH)D) and details of the measurement method. Include descriptive statistics of vitamin D levels in the sample. Report whether this outcome was defined as continuous or categorical (e.g., very deficient, deficient, adequate). Standardise the distribution to enable comparison across populations, which may differ significantly in mean or range of vitamin D.
Genetics	Report clearly on chosen genetic factor. Researchers are also encouraged to aim to replicate previous findings where possible.
Environment	Use independent UV radiation data from sources such as NASA or Google Earth alongside personal sun exposure habits. Quantitative sun exposure data allows comparison across studies.
Interaction	Report clearly on the model parameters and interaction term(s) as well as the effect estimates and statistical significance of G, E, and GxE. Include the reasoning for choosing the model and assumptions made. Report GxE results even if not significant.
Sample	Include descriptive statistics of the sample such as age and sex. While the field broadly would benefit from larger and more ethnically and geographically diverse samples, this may not be possible for individual studies. Where possible, researchers should consider sampling underrepresented populations to broaden ancestry coverage within vitamin D research. Report on the ethnicity and geography of the sampled population and any analysis of population structure.
Covariates	Evaluate known covariates associated with vitamin D—age, sex, and BMI. Consider other covariates such as season of blood draw, ethnicity, skin colour, and vitamin D supplement intake.

## Data Availability

The study did not report any new data.

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
