# Peer review of "Gene-Environment Interactions in Vitamin D Status and Sun Exposure: A Systematic Review with Recommendations for Future Research"

_nutrients, 2022, doi:10.3390/nu14132735_

Round 1
Reviewer 1 Report
Title and abstract: It seems the article mainly focuses on “sun exposure” So I believe it would be better to highlight “sun exposure” in the title and abstract. In the current title and abstract, “environment” seems to covered the “sun exposure.” However, as the authors summarized, there could be many other environmental factors other than sun exposure.
Table 2. Need more clear separation between each rows. The current one looks a little crowded and mingled together, especially between Livingstone 2017 and Shao 2018- very hard to read and distinguish which row is which
Line 138. Since sun exposure is closely related to season, I wonder how using season as a covariate would affect relevant studies, and eventually your analysis.
Line 322: would it be rational to include the “expanding ancestry coverages” as one of the recommendations in Table 3? If so please consider.
Author Response
Response to Reviewer 1 Comments
Point 1: Title and abstract: It seems the article mainly focuses on “sun exposure” So I believe it would be better to highlight “sun exposure” in the title and abstract. In the current title and abstract, “environment” seems to covered the “sun exposure.” However, as the authors summarized, there could be many other environmental factors other than sun exposure.
Response 1: We originally used “environment” because we searched for environment and did not specify “sun exposure.” However, as the reviewer advises, we have updated our title and abstract to accurately reflect that “sun exposure” is the only environmental exposure currently available in the literature on GxE in vitamin D. The title has been changed to emphasise sun exposure (new title: “Gene-environment interactions in vitamin D status and sun exposure: a systematic review with recommendations for future research”). The abstract has also been edited accordingly.
Point 2: Table 2. Need more clear separation between each rows. The current one looks a little crowded and mingled together, especially between Livingstone 2017 and Shao 2018- very hard to read and distinguish which row is which
Response 2: Table 2 has been formatted in LaTeX to be more readable.
Point 3: Line 138. Since sun exposure is closely related to season, I wonder how using season as a covariate would affect relevant studies, and eventually your analysis.
Response 3: We have focused, in our manuscript, on the use of independent UV data as the environmental exposure instead of season. However, it is possible, as the reviewer suggests, to use season as a covariate in addition to sun exposure as the environmental exposure. Section 4.3 Choice of environmental exposure discusses the choice of season compared to independent UV radiation as the exposure variable. See lines 235 - 244 highlighting that “because of the large UVB dose variability within any given season, and because of the large UVB dose variability between locations but during the same time of the year, season of blood draw is a poor proxy of UVB exposure.” We have added to this point the use of season as a covariate. The effect of season may relate to individuals’ behaviour patterns across the year. For example, if people are more likely to spend time outside during the summer then the difference between ‘high sun exposure’ in summer and ‘low sun exposure’ in winter is compounded by the time spent outdoors. The usefulness of this variable will depend on the location and data availability of the study but we agree that it should be evaluated where possible.
Point 4: Line 322: would it be rational to include the “expanding ancestry coverages” as one of the recommendations in Table 3? If so please consider.
Response 4: This is an important recommendation and key takeaway from the review of the available research. We have incorporated it into Table 3 by making the recommendation to select undersampled ancestries explicit: “Where possible, researchers should consider sampling underrepresented populations to broaden ancestry coverage within vitamin D research.”
Reviewer 2 Report
This is a very interesting systematic review describing the evidence on the effects of gene-environment interactions on vitamin D status. The paper is well-written, below are my comments to the authors:
Major
1. My main concern is that the search was conducted over a year ago, so must be updated to ensure all relevant studies are included in this systematic review.
2. The health condition of each study population must be explained and added to Table 1. Also, skin color can be a determinant of vitamin D status as people with darker skin (type 4 to 6 on Fitzpatrick scale) tend to have lower serum 25(OH)D. These two affect generalizability of study findings and should be considered in discussing the inconsistencies found in available studies.
Author Response
Response to Reviewer 2 Comments
This is a very interesting systematic review describing the evidence on the effects of gene-environment interactions on vitamin D status. The paper is well-written, below are my comments to the authors:
Major
- My main concern is that the search was conducted over a year ago, so must be updated to ensure all relevant studies are included in this systematic review.
Response 1: We thank the reviewer for this valuable revision. The systematic review should reflect the available literature as close to the time of its publication as possible. We repeated the systematic search of Embase, Medline (Ovid), and Web of Science (Core Collection) (as outlined in 2.1 Search strategy) from January 2021 to current, i.e. 15 June 2022. 68 records were identified in total (25, 24 and 19, respectively). 41 records remained after removing duplicates. Similarly to the original search, many of the records were excluded because they were reviews or because they examined vitamin D genes or vitamin D as the exposure rather than the outcome (for example, GxE in lupus erythematous DOI: 10.1111/joim.13448 or GxE in renal carcinoma DOI: 10.1186/s41021-021-00185-3). The title and abstract screening identified only 1 record that was potentially relevant, but it was excluded after a full text review (the text was a conference abstract focusing on genetic associations rather than interactions). So, no additional eligible studies were found. We have updated the date in the methods section 2.1 Search strategy from February 2021 to 15 June 2022 to reflect the extended search period as well as corrected and updated the PRISMA flow diagram to include the additional titles screened.
2. The health condition of each study population must be explained and added to Table 1. Also, skin color can be a determinant of vitamin D status as people with darker skin (type 4 to 6 on Fitzpatrick scale) tend to have lower serum 25(OH)D. These two affect generalizability of study findings and should be considered in discussing the inconsistencies found in available studies.
Response 2: The sample description in Table 1 has been updated to include the health information of each study population, where available. While some of the studies sampled from disease or population-based cohorts, none targeted a specific disease or examined association with a specific disease outcome other than vitamin D status.
Beyond reporting ancestry, unfortunately, none of the included studies measured skin type or colour or pigmentation. As this is a very pertinent issue, section 4.5 Methodological implications in the Discussion and Table 3 have been updated to include a recommendation on skin colour.
Round 2
Reviewer 2 Report
Thank you for addressing my comments and best of luck with your research!